# REWIRING WITH POSITIONAL ENCODINGS FOR GNNS

## ABSTRACT

Several recent works use positional encodings to extend the receptive fields of graph neural network (GNN) layers equipped with attention mechanisms. These techniques, however, extend receptive fields to the complete graph, at substantial computational cost and risking a change in the inductive biases of conventional GNNs, or require complex architecture adjustments. As a conservative alternative, we use positional encodings to expand receptive fields to $r$-hop neighborhoods. More specifically, our method augments the input graph with additional nodes/edges and uses positional encodings as node and/or edge features. We thus modify graphs before inputting them to a downstream GNN model, instead of modifying the model itself. This makes our method model-agnostic, i.e. compatible with any existing GNN architectures. We also provide examples of positional encodings that are lossless with a one-to-one map between the original and the modified graphs. We demonstrate that extending receptive fields via positional encodings and a virtual fully-connected node significantly improves GNN performance and alleviates over-squashing using small $r$. We obtain improvements on a variety of models and datasets, and reach state-of-the-art performance using traditional GNNs or graph Transformers.

## 1 INTRODUCTION

GNN layers typically embed each node of a graph as a function of its neighbors' (1-ring's) embeddings from the previous layer; that is, the *receptive field* of each node is its 1-hop neighborhood. Hence, at least $r$ stacked GNN layers are needed for nodes to get information about their $r$-hop neighborhoods. Barceló et al. (2020) and Alon and Yahav (2021) identify two broad limitations associated with this structure: *under-reaching* occurs when the number of layers is insufficient to communicate information between distant vertices, while *over-squashing* occurs when certain edges act as bottlenecks for information flow.

Inspired by the success of the Transformer in natural language processing (Vaswani et al., 2017), recent methods expand node receptive fields to the whole graph (Dwivedi and Bresson, 2021; Ying et al., 2021). Since they effectively replace the topology of the graph with that of a complete graph, these works propose *positional encodings* that communicate the connectivity of the input graph as node or edge features. As these methods operate on fully-connected graphs, the computational cost of each layer is quadratic in the number of nodes, obliterating the sparsity afforded by conventional 1-ring based architectures. Moreover, the success of the 1-ring GNNs suggests that local feature aggregation is a useful inductive bias, which has to be learned when the receptive field is the whole graph, leading to slow and sensitive training.

In this paper, we expand receptive fields from 1-ring neighborhoods to $r$-ring neighborhoods, where $r$ ranges from 1 (typical GNNs) to $R$, the diameter of the graph (fully-connected). That is, we augment a graph with edges between each node and all others within distance $r$ in the input topology. We show that performance is significantly improved using fairly small $r$ and carefully-chosen positional encodings annotating this augmented graph. This simple but effective approach can be combined with any GNN.

**Contributions.** We apply GNN architectures to augmented graphs connecting vertices to their peers of distance $\leq r$. Our contributions are as follows: (i) We increase receptive fields using a modified graph with positional encodings as edge and node features. (ii) We compare $r$-hop positional encodings on the augmented graph, specifically lengths of shortest paths, spectral computations, and

powers of the graph adjacency matrix. (iii) We demonstrate that relatively small $r$-hop neighborhoods sufficiently increase performance across models and that performance degrades in the fully-connected setting.

## 2 RELATED WORK

The Transformer has permeated deep learning (Vaswani et al., 2017), with state-of-the-art performance in NLP (Devlin et al., 2018), vision (Parmar et al., 2018), and genomics (Zaheer et al., 2020). Its core components include multi-head attention, an expanded receptive field, positional encodings, and a CLS-token (virtual global source and sink nodes). Several works adapt these constructions to GNNs. For example, the Graph Attention Network (GAT) performs attention over the neighborhood of each node, but does not generalize multi-head attention using positional encodings (Veličković et al., 2018). Recent works use Laplacian spectra, node degrees, and shortest-path lengths as positional encodings to expand attention to all nodes (Kreuzer et al., 2021; Dwivedi and Bresson, 2021; Rong et al., 2020; Ying et al., 2021). Several works also adapt attention mechanisms to GNNs (Yun et al., 2019; Cai and Lam, 2019; Hu et al., 2020; Baek et al., 2021; Veličković et al., 2018; Wang et al., 2021b; Zhang et al., 2020; Shi et al., 2021).

Path and distance information has been incorporated into GNNs more generally. Yang et al. (2019) introduce the Shortest Path Graph Attention Network (SPAGAN), whose layers incorporate path-based attention via shortest paths between a center node and distant neighbors, using an involved hierarchical path aggregation method to aggregate a feature for each node. Like us, SPAGAN introduces the $\leq k$-hop neighbors around the center node as a hyperparameter; their model, however, has hyperparameters controlling path sampling. Beyond SPAGAN, Chen et al. (2019) concatenate node features, edge features, distances, and ring flags to compute attention probabilities. Li et al. (2020) show that distance encodings (i.e., one-hot feature of distance as an extra node attribute) obtain more expressive power than the 1-Weisfeiler-Lehman test. Graph-BERT introduces multiple positional encodings to apply Transformers to graphs and operates on sampled subgraphs to handle large graphs (Zhang et al., 2020). Yang et al. (2019) introduce the Graph Transformer Network (GTN) for learning a new graph structure, which identifies "meta-paths" and multi-hop connections to learn node representations. Wang et al. (2021a) introduce Multi-hop Attention Graph Neural Network (MAGNA) that uses diffusion to extend attention to multi-hop connections. Frankel et al. (2021) extend GAT attention to a stochastically-sampled neighborhood of neighbors within 5-hops of the central node. Isufi et al. (2020) introduce EdgeNets, which enable flexible multi-hop diffusion. Luan et al. (2019) generalizes spectral graph convolution and GCN in block Krylov subspace forms.

Each layer of our GNN attends to the $r$-hop neighborhood around each node. Unlike SPAGAN and Graph-BERT, our method is model agnostic and does not perform sampling, avoiding their sampling-ratio and number-of-iterations hyperparameters. Unlike GTN, we do not restrict to a particular graph structure. Broadly, our approach does not require architecture or optimization changes. Thus, our work also joins a trend of decoupling the input graph from the graph used for information propagation (Veličković, 2022). For scalability, Hamilton et al. (2017) sample from a node's local neighborhood to generate embeddings and aggregate features, while Zhang et al. (2018) sample to deal with topological noise. Rossi et al. (2020) introduce Scalable Inception Graph Neural Networks (SIGN), which avoid sampling by precomputing convolutional filters. Kipf and Welling (2017) pre-process diffusion on graphs for efficient training. Topping et al. (2021) use graph curvature to rewire graphs and combat over-squashing and bottlenecks.

In contrast, our work does not use diffusion, curvature, or sampling, but expands receptive fields via Transformer-inspired positional encodings. In this sense, we avoid the inductive biases from pre-defined notions of diffusion and curvature, and since we do not remove connectivity, injective lossless changes are easy to obtain.

## 3 PRELIMINARIES AND DESIGN

Let $G = (V, E, f_v, f_e)$ denote a graph with nodes $V \subset \mathbb{N}_0$ and edges $E \subseteq V \times V$, and let $\mathcal{G}$ be the set of graphs. For each graph, let functions $f_v : V \to \mathbb{R}^{d_v}$ and $f_e : E \to \mathbb{R}^{d_e}$ denote node and edge features, respectively. We consider learning on graphs, specially node classification and graph classification. At inference, the input is a graph $G$. For node classification, the task is to predict

a node label $l_v(v) \in \mathbb{R}$ for each vertex $v \in V$. Using the node labels, the homophily of a graph is defined as the fraction of edges that connect nodes with the same labels (Ma et al., 2022). For graph classification, the task is to predict a label $l_G \in \mathbb{R}$ for the entire graph $G$.

Given the tasks above, GNN architectures typically ingest a graph $G = (V, E, f_v, f_e)$ and output either a label or a per-node feature. One can view these as an abstraction; e.g. a GNN for graph classification is a map $F_\theta : \mathcal{G} \to \mathbb{R}^n$ with learnable parameters $\theta$. These architectures vary in terms of how they implement $F_\theta$. Some key examples include the following: (i) Spatial models (Kipf and Welling, 2017) use the graph directly, computing node representations in each layer by aggregating representations of a node and its neighbors (1-ring). (ii) Spectral models (Bruna et al., 2014) use the eigendecomposition of the graph Laplacian to perform spectral convolution. (iii) Diffusion models (Wang et al., 2021a; Klicpera et al., 2019) use weighted sums of powers of the adjacency matrix to incorporate larger neighborhoods ($r$-hops). (iv) In Transformers (Kreuzer et al., 2021; Dwivedi and Bresson, 2021; Rong et al., 2020; Ying et al., 2021), each node forms a new representation by self-attention over the complete graph ($R$-hop neighborhood) using positional encodings. These approaches incorporate useful inductive biases while remaining flexible enough to learn from data.

Spatial models have been extremely successful, but recent work shows that they struggle with under-reaching and over-squashing (Alon and Yahav, 2021). Spectral approaches share similar convolutional bias as spatial models and face related problems (Kipf and Welling, 2017). On the other hand, Transformers with complete attention and diffusion aim to alleviate the shortcomings of spatial models and show promising results. Due to complete attention, Transformers carry little inductive bias but are also computationally expensive. Diffusion explicitly incorporates the inductive bias that distant nodes should be weighted less in message aggregation; limiting its breadth of applicability.

We alleviate under-reaching and over-squashing while avoiding the computational load of complete attention by incorporating a more general proximity bias than diffusion without committing to a specific model. Our method is built on the observation that $F_\theta$ can be trained to ingest modified versions of the original graph that better communicate structure and connectivity. Hence, we add new edges, nodes, and features to the input graph. To still convey the original topology of the input graph, we add positional encodings. More formally, we design functions $g : \mathcal{G} \to \mathcal{G}$ that modify graphs and give features to the new nodes and edges. These functions can be prepended to any GNN $F_\theta : \mathcal{G} \to \mathbb{R}^n$ as $F_\theta \circ g : \mathcal{G} \to \mathbb{R}^n$.

The following are desiderata informing our design of $g$: (i) ability to capture the original graph, (ii) ability to incorporate long-range connections, (iii) computational efficiency, and (iv) minimal and flexible locality bias. By using positional encodings and maintaining the original graph $G$ as a subgraph of the modified graph, we capture the original graph in our modified input (Section 4.2.1). By expanding the receptive field around each node to $r$-hop neighborhoods we reduce computational load relative to complete-graph attention, with limited inductive bias stemming from proximity. Additionally, expanded receptive fields alleviate under-reaching and over-squashing (Section 6.1).

# 4 APPROACH

We modify graphs before inputting them to a downstream GNN model, instead of modifying the model itself. Our approach does not remove edges or nodes in the original graph but only adds elements. Given input $G = (V, E, f_v, f_e)$, we create a new graph $G' = (V', E', f_v', f_e')$ such that $G$ is a subgraph of $G'$. Expanded receptive fields are achieved in $G'$ by adding edges decorated with positional encodings as node or edge attributes; we also add a fully-connected CLS node. $G'$ is still a graph with node and edge attributes to which we may apply any GNN. This process is represented by a function $g : \mathcal{G} \to \mathcal{G}$. We decompose the construction of $g$ into topological rewiring and positional encoding, detailed below. In a slight abuse of notation, we will subsequently use $\mathcal{G}$ to denote only the subset of graphs relevant to a given machine learning problem. For example, for graph regression on molecules, $\mathcal{G}$ denotes molecule graphs, with atoms as nodes and bonds as edges.

## 4.1 TOPOLOGICAL REWIRING

We modify the input graph $G$ to generate $G'$ in two steps:

**Expanded receptive field.** Given a graph $G = (V, E, f_v, f_e) \in \mathcal{G}$ and a positive integer $r \in \mathbb{N}_+$, we add edges between all nodes within $r$ hops of each other in $G$ to create $G'_r = (V, E', f'_v, f'_e)$. If $G$ is annotated with edge features, we assign to each edge in $E' \backslash E$ an appropriate constant feature $C_e$.

**CLS node.** Following Gilmer et al. (2017), we also include a "CLS"—or classification—node to our graph connected to all others. We follow this procedure: Given a graph $G$, we (i) initialize a new graph $G' = (V', E', f'_v, f'_e) = G$, (ii) add a new node $v_{\text{CLS}}$ to $V'$, and (iii) set $f'_v(v_{\text{CLS}}) \coloneqq C_v$ for a constant $C_v$. Finally, we set $E' \coloneqq E \cup \bigcup_{v \in V} \{(v_{\text{CLS}}, v), (v, v_{\text{CLS}})\}$, with $f'_e((v_{\text{CLS}}, v)) = f'_e((v, v_{\text{CLS}})) \coloneqq C_e$, where $C_e$ is defined above.

## 4.2 POSITIONAL ENCODINGS

Given only the connectivity of a rewired graph $G'_r = (V', E', f'_v, f'_e)$ from the two-step procedure above, it may not be possible to recover the connectivity of the original graph $G = (V, E, f_v, f_e)$. In the extreme, when $r$ is large and $G$ is connected, $G'_r$ could become fully-connected, meaning that all topology is lost—removing the central cue for graph-based learning. To combat this, we encode the original topology of $G$ into $G'_r$ via *positional encodings*, which are node and/or edge features. We consider several positional encoding functions for edges $p_e : \mathcal{G} \times V' \times V' \to \mathbb{R}^n$ or nodes $p_v : \mathcal{G} \times V' \to \mathbb{R}^n$, appending the output of $p_e$ as edge or $p_v$ as node features to $G'_r$. Section 4.2.1 lays out properties to compare choices of $p_e$ and/or $p_v$. Then, Section 4.2.2 provides concrete positional encodings compared in our experiments that trade off between the properties we lay out.

### 4.2.1 PROPERTIES OF POSITIONAL ENCODINGS

There are countless ways to encode the subgraph topology of $G$ within $G'$ in vertex features $p_v$ or edge features $p_e$. Below, we state a few properties we can check to give a framework for comparing the capabilities and biases of possible choices.

**Lossless encoding.** While a GNN can ignore information in input $G'$, it cannot reconstruct information that has been lost in constructing $G'$ from $G$. Yet, there can be benefits in forgetting information, e.g. when dealing with noisy graphs or incorporating a stronger inductive bias (Rossi et al., 2020; Klicpera et al., 2019). That said, a simple property to check for $G'$ equipped with positional encoding features $p_e, p_v$ is whether we can recover $G$ from this information, that is, whether our encoding is *lossless* (or *non-invasive*). As long as it is possible to identify $G$ within $g(G)$, $g$ is an injection and non-invasive. Hence, a sufficient condition for lossless positional encodings is as follows: If all edges in $G'$ have unique positional encodings, then $g : \mathcal{G} \to \mathcal{G}$ is a bijection. One way to achieve this condition is to use an additional edge feature that is unique to the 1-ring.

**Discriminative power.** Following work investigating the discriminative power of GNNs (Xu et al., 2019; Brüel Gabrielsson, 2020), Ying et al. (2021) showed that expanded receptive fields together with shortest-path positional encodings are strictly more powerful than the 1-Weisfeiler-Lehman (WL) test and hence more powerful than 1-hop vanilla spatial GNN models (Xu et al., 2019). The combination of increased receptive fields, positional encodings, and choice of subsequent GNN models determines discriminative power. In fact, it follows from (Ying et al., 2021) that the positional encodings presented below together with an increased receptive field $r > 1$ and a vanilla spatial GNN model are strictly more powerful than the 1-WL test.

**Computational time.** Positional encodings may come at substantial computational cost when working with $r$-hop neighborhoods. The cost of computing positional encodings affects total inference time, which may be relevant in some learning settings. However, in our setting the runtime of computing positional encodings is an order of magnitude less than the subsequent inference time, and in our implementation the asymptotic runtimes of computing the positional encodings are the same. See Appendix E.

**Local vs. global.** The positional encoding of a vertex or edge can be local, meaning it incorporates information from a limited-sized neighborhood in $G$, or global, in which case adding or removing a node anywhere in $G$ could affect all the positional encodings.

**Inductive bias.** Our positional encodings can bias the results of the learning procedure, effectively communicating to the downstream GNN which properties of $G$ and $G'$ are particularly important for learning. Without positional encodings, our model would induce a bias stating that distances $< r$ in our graph are insignificant. More subtly, suppose $\ell$ is the distance (of length $\leq r$) between two

nodes in $G$ corresponding to a new edge in $E'$. Using $\ell$ directly as positional encoding rather than a decaying function, e.g. $e^{-\alpha\ell}$, makes it easier or harder (resp.) to distinguish long distances in $G$.

A related consideration involves whether our model can imitate the inductive bias of past work. For example, graph diffusion has been used to incorporate multi-hop connections into GNNs using fixed weights (Wang et al., 2021a). We can ask whether our positional encodings on $G'$ are sufficient to learn to imitate the behavior of a prescribed multi-hop model on $G$, e.g. whether a layer of our GNN applied to $G'$ can capture multi-hop diffusion along $G$.

**Over-squashing and under-reaching.** Section 6.1 demonstrates, via the NeighborsMatch problem (Alon and Yahav, 2021), that increased receptive fields as well as the CLS-node alleviate over-squashing; however, this toy problem is concerned with matching node attributes and not with graph topology. We want positional encodings that alleviate over-squashing in the sense that it enables effective information propagation for the task at hand. Our experiments showing that expanded receptive fields alleviate the over-squashing problem and that the best performing positional encoding varies across datasets showcase this. Additionally, our experiments on the discriminative power of positional encodings in Appendix D further help discern the different options.

### 4.2.2 POSITIONAL ENCODING OPTIONS

Taking the properties above into consideration, we now give a few options for positional encodings below, compared empirically in Section 6.

**Shortest path.** For any edge $e \in G'_r$, the *shortest-path positional encoding* takes $p_e \in \{0, 1, \ldots, r\}$ to be the integer length of the shortest path in $G$ between the corresponding nodes of $E$. These embeddings are lossless because $G$ is the subgraph of $g(G)$ with $p_e = 1$. They also are free to compute given our construction of $G'_r$ from $G$. But, multiple vertices in the $r$-neighborhood of a vertex in $V$ could have the same positional encoding in $V'$, and shortest path lengths are insufficient to capture complex inductive biases of multi-hop GNNs like diffusion over large neighborhoods. Shortest-path positional encoding was previously used by Ying et al. (2021), for extending $G$ to a fully-connected graph, but they did not consider smaller $r$ values.

**Spectral embedding.** Laplacian eigenvectors embed graph vertices into Euclidean space, providing per-vertex features that capture multi-scale graph structure. They are defined by factorizing the graph Laplacian matrix, $\Delta = I - D^{-1/2}AD^{-1/2}$, where $D$ is the degree matrix and $A$ is the adjacency matrix. We call the result a *spectral positional embedding*. We can use the $q$ smallest non-trivial Laplacian eigenvectors of $G$ as a node-based positional encoding $p_v : V' \to \mathbb{R}^q$. Following Dwivedi et al. (2020), since these eigenvectors are known only up to a sign, we randomly flip the sign during training. Prior work consider Laplacian eigenvectors as additional node features without topological rewiring (Dwivedi et al., 2020).

Spectral positional encodings do not necessarily make $g$ injective. Even when $q = |V|$, this encoding fails to distinguish isospectral graphs (Von Collatz and Sinogowitz, 1957), but these are rarely encountered in practice. On the other hand, spectral signatures are common for graph matching and other tasks. Moreover, unlike the remaining features in this section, spectral positional encodings capture global information about $G$ rather than only $r$-neighborhoods. Finally, we note that the diffusion equation for graphs can be written as $u_t = -\Delta u$; this graph PDE can be solved in closed-form given the eigenvectors and eigenvalues of $\Delta$. Hence, given the spectral embedding of $G$ in $G'$, we can simulate diffusion-based multi-hop GNN architectures up to spectral truncation.

**Powers of the adjacency matrix.** Our final option for positional encoding generalizes the shortest path encoding and can capture the inductive biases of diffusion-based GNNs. The entry at position $(i, j)$ of the $k$-th power $A^k$ of the adjacency matrix $A$ of graph $G$ gives the number of paths of length $k$ between node $i$ and $j$ in $G$. Concatenating the powers from $k = 1, \ldots, r$, we get for each edge $e$ in $G'$ an integer vector $p_e \in \mathbb{N}^r$ giving the *powers of the adjacency matrix positional encoding*.

This embedding can be used to recover the shortest-path embedding. This adjacency-derived embedding can also generalize the inductive bias of diffusion-based multi-hops GNNs. In particular, diffusion aggregation weights are often approximated using a Taylor series, $\mathcal{W} = \sum_{i=0}^{\infty} \theta_i A^i \approx \sum_{i=0}^{r} \theta_i A^i := W$, where $\theta_i$ are a prescribed decaying sequence ($\theta_i > \theta_{i+1}$). The entries of $W$ above can be computed linearly from the adjacency-powers positional encoding. Hence, it is strictly more general than using prescribed diffusion-based aggregation weights on $G$.

**Lossless encodings.** The previously discussed lossless-encoding properties of our graph rewiring method are accomplished by two of the above-mentioned positional encodings:

**Proposition 1.** *Shortest-path and adjacency matrix positional encodings yield lossless rewirings.*

*Proof.* Recovering the original graph $G = (V, E)$ from the rewired graph $G' = (V, E')$ is almost trivial. With the shortest-path position encoding the original graph can be recovered via $E = \{e | e \in E', p_e = 1\}$ and for powers-of-the-adjacency-matrix encodings via $E = \{e | e \in E', (p_e)_1 = 1\}$. $\square$

## 5 IMPLEMENTATION DETAILS

Our method is compatible with most GNN architectures. Here we adopt GatedGCN (Bresson and Laurent, 2018), MoNet (Monti et al., 2017), and an implementation of the Transformer (Vaswani et al., 2017); see Appendix B for details. For each model, we consider graph rewiring with a different $r$-hop receptive field around each node, and compare with and without the CLS-node, as well as the three positional encodings introduced in Section 4.2.2.

**Input and readout layers.** Typically, GNNs on a graph $G = (V, E, f_v, f_e)$ first embed node features $f_v$ and edge features $f_e$ through a small feed-forward network (FFN) *input layer*. When incorporating positional encodings per edge/node, we embed using a small FFN and add them at this input layer. After this layer, it updates node and edge representations through successive applications of GNN layers. Lastly, a *readout layer* is applied to the last GNN layer $L$. For node classification, it is typically a FFN applied to each node feature $h_i^L$. For graph classification, it is typically an FFN applied to the mean or sum aggregation of all node features $h^L$. For graph classification and when using the CLS-node, we aggregate by applying the FFN to the CLS-node's features in the last layer.

## 6 EXPERIMENTS

We evaluate performance on six benchmark graph datasets: ZINC, AQSOL, PATTERN, CLUSTER, MNIST, and CIFAR10 from (Dwivedi et al., 2020). The benchmark includes a training time limit of 12 hours; we use similar compute to their work via a single TeslaV100 GPU. Training also stops if for a certain number of epochs the validation loss does not improve (Dwivedi et al., 2020). Thus, our experiments consider the ease of training and efficient use of compute. For the first two datasets, we run GatedGCN, MoNet, and Transformer to show that rewiring and positional encoding work for different models; for the other datasets we run only GatedGCN to focus on the effects of receptive field size, the CLS node, and positional encodings. For all datasets, we run with increasing receptive fields, with different positional encodings, and with or without the CLS-node. In the tables, *density* is the average of the densities (defined as the ratio $|E|/|V|^2$) of each graph in the dataset rewired to the respective receptive field size. See Appendix A for details.

Table 1 compares our best results with other top performing methods and models. All our top performing models come from the GatedGCN, although the Transformer performs comparably; however, the Transformer was harder to train—see Appendix B. MoNet performs worse but still sees significant improvements from our approach. Our GatedGCN implementation was taken from the same work (Dwivedi et al., 2020) that introduced the benchmarks and code that we use. Thus, hyperparameters might be better adapted to the GatedGCN. This highlights the benefits of our model-agnostic approach, which allows us to pick the best models from Dwivedi et al. (2020) and combine them with our methods. Our approach with 100K parameters achieves state-of-the-art on all datasets among models with 100K parameters and even outperforms 500K-parameter models.

**ZINC, Graph Regression.** ZINC consists of molecular graphs and the task is graph property regression for constrained solubility. Each ZINC molecule is represented as a graph of atoms with nodes and bonds as edges. In Table 2 we present results for $r$ from 1 to 10. The *density* column shows that these graphs are sparse and that the number of edges increases almost linearly as the receptive field $r$ is increased. Performance across all settings noticeably improves when increasing $r$ above 1. Top performance is achieved with the CLS-node and powers-of-the-adjacency positional encoding at $r = 4$, and at 52% of the edges and compute compared to complete attention. When using the CLS node and/or spectral positional encodings, top performance generally occurs at lower $r$, which is likely due to the global nature of these changes to the graphs. The GatedGCN and

Transformer perform comparably for the same settings, with a slight edge to the GatedGCN. The two models show the same performance trends between settings, i.e., both increased receptive fields and the CLS-node boost performance. Further, Ying et al. (2021) include a performance of 0.123 on ZINC with their Graphormer(500K), i.e., a Transformer with positional encodings and complete attention. However, their training is capped at 10,000 epochs while ours is capped at 1,000 epochs; training their Graphormer(500K) with same restrictions leads to a score of 0.26 on ZINC.

**AQSOL, Graph Regression.** AQSOL consists of the same types of molecular graphs as ZINC. The densities of AQSOL graphs are slightly higher than those of ZINC. For all settings not including CLS-node or spectral positional encodings, performance improves significantly when increasing $r$ above 1 (see Table 3); in these settings, better performing $r$ are larger than for ZINC. However, when including CLS node or spectral positional encodings, performance changes much less across different $r$. This indicates the importance of some form of global bias on this dataset. At least one of larger $r$ values, spectral positional encoding, or the CLS-token is required to provide the global bias, but the effect of them differs slightly across the two models. GatedGCN performs significantly better, and larger $r$-values still boosts performance when combined with the CLS-token for MoNet, but not for GatedGCN. MoNet uses a Bayesian Gaussian Mixture Model (Dempster et al., 1977) and since MoNet was not constructed with edge-features in mind, we simply add edge embeddings to the attention coefficients. Not surprisingly, this points to the importance of including edge features for optimal use of expanded receptive fields and positional encodings.

**CLUSTER, Node Classification.** CLUSTER is a node classification dataset generated using a stochastic block model (SBM). The task is to assign a cluster label to each node. There is a total of 6 cluster labels and the average homophily is 0.34. CLUSTER graphs do not have edge features. Table 4 gives results for $r$-hop neighborhoods from 1 to 3. As can be seen in the density column, at $r = 3$ all graphs are fully connected, and more than 99% of them are fully connected at $r = 2$. Hence, these graphs are dense. Significant improvements are achieved by increasing $r$ for all but the spectral positional encoding (again showcasing its global properties), which together with the CLS node perform competitively at $r = 1$. The CLS node is helpful overall, especially at $r = 1$. The GatedGCN and Transformer perform comparably for all but the spectral positional encodings where the Transformer breaks down. We found that this breakdown was due to the placement of batch normalization, discussed in Appendix B.1.

**PATTERN, Node Classification.** The PATTERN dataset is also generated using a SBM model, and has an average homophily of 0.66. The task is to classify the nodes into two communities and graphs have no edge features. Table 5 shows results for $r$-hops from 1 to 3. Similarly to CLUSTER, the density column shows that the graphs are dense. Significant improvements are achieved by increasing $r > 1$ and/or using the CLS-node. Performance generally decreases at $r = 3$. Similarly to CLUSTER, the CLS-node helps at $r = 1$, but for both CLUSTER and PATTERN, top performing model comes from a larger $r > 1$ without the CLS-node, suggesting that trade-offs exist between CLS-node and increased receptive fields. Compared to CLUSTER, our approach shows less performance boost for PATTERN, which lead us to hypothesize that our approach is more helpful for graphs with low homophily which we investigate further in Appendix F.

**MNIST, Graph Classification.** MNIST is an image classification dataset converted into super-pixel graphs, where each node's feature includes super-pixel coordinates and intensity. The images are of handwritten digits, and the task is to classify the digit. Table 6 summarizes results for $r$ from 1 to 3. Not all graphs are fully connected at $r = 3$, but training at $r = 4$ exceeds our memory limit. Noticeable performance gains are achieved at $r = 2$, but performance generally decreases at $r = 3$. The CLS-node consistently improves performance at $r = 1$ but not otherwise, indicating that the CLS-node and increased $r$-size have subsumed effects.

**CIFAR10, Graph Classification.** CIFAR10 is an image classification dataset converted into super-pixel graphs, where each node's features are the super-pixel coordinates and intensity. The images consist of ten natural motifs, and the task is to classify the motif, e.g., dog, ship, or airplane. Table 7 provides results for $r$ from 1 to 3. Not all graphs are fully connected at $r = 3$, but training at $r = 4$ led to out-of-memory issues. Top performing versions are all at $r = 1$, and performance degrades for $r > 1$. As with MNIST, the CLS-node only improves performance at $r = 1$, again indicating its shared (subsumed) effects with increased $r$-sizes.

Table 1: Benchmarking. Higher is better for all but for ZINC and AQSOL where lower is better. Benchmarks can be found in Dwivedi et al. (2020); Corso et al. (2020); Bouritsas et al. (2020); Dwivedi and Bresson (2021). The benchmarks (Dwivedi et al., 2020) and corresponding leaderboard have 100K and 500K parameter entries. OOM is short for out-of-memory errors.

| Datasets: | PATTERN | CLUSTER | MNIST | CIFAR10 | ZINC | AQSOL |
|---|---|---|---|---|---|---|
| task: | node class. | node class. | graph class. | graph class. | graph reg. | graph reg. |
| # graphs: | 14000 | 12000 | 70000 | 60000 | 12000 | 9823 |
| Avg # nodes: | 117.47 | 117.20 | 70.57 | 117.63 | 23.16 | 17.57 |
| Avg # edges: | 4749.15 | 4301.72 | 564.53 | 941.07 | 49.83 | 35.76 |
| MoNet(100K) | 85.482±0.037 | 58.064±0.131 | 90.805±0.032 | 54.655±0.518 | 0.397±0.010 | 1.395±0.027 |
| GAT(100K) | 75.824±1.823 | 57.732±0.323 | 95.535±0.205 | 64.223±0.455 | 0.475±0.007 | 1.441±0.023 |
| GraphSage(100K) | 50.516±0.001 | 50.454±0.145 | 97.312±0.097 | 65.767±0.308 | 0.468±0.003 | 1.431±0.010 |
| GIN(100K) | 85.590±0.011 | 58.384±0.236 | 96.485±0.252 | 55.255±1.527 | 0.387±0.015 | 1.894±0.024 |
| PNA(100K) | **86.730±0.050** | 63.020±0.262 | 97.940±0.120 | 70.350±0.630 | 0.188±0.004 | 1.083±0.011 |
| GatedGCN(100K) | 84.480±0.122 | 60.404±0.419 | 97.340±0.143 | 67.312±0.311 | 0.328±0.003 | 1.295±0.016 |
| GatedGCN-PE/E(500K) | 86.363±0.127 | 74.088±0.344 | OOM | OOM | 0.214±0.006 | 0.996±0.008 |
| GraphTransformer(500K) | 54.941±3.739 | 27.121±8.471 | OOM | OOM | 0.598±0.049 | 1.110±0.010 |
| Ours(100K) | **86.757±0.031** | **77.575±0.149** | **98.743±0.062** | **73.808±0.193** | **0.143±0.006** | **0.920±0.009** |

Table 2: Increasing $r$ on ZINC/molecules 100K parameters.

| type: | density | trans-adj | trans-adj-cls | trans-short | trans-short-cls | trans-lp | trans-lp-cls | gcn-adj | gcn-adj-cls | gcn-short | gcn-short-cls | gcn-lp | gcn-lp-cls |
|---|---|---|---|---|---|---|---|---|---|---|---|---|---|
| r=1 | .14 | 0.341±0.024 | 0.289±0.012 | 0.346±0.022 | 0.298±0.012 | 0.293±0.044 | 0.257±0.036 | 0.329±0.023 | 0.287±0.010 | 0.326±0.024 | 0.265±0.043 | 0.291±0.029 | 0.274±0.027 |
| r=2 | .27 | 0.297±0.019 | 0.234±0.021 | 0.295±0.030 | 0.220±0.040 | **0.263±0.024** | 0.253±0.030 | 0.265±0.021 | 0.198±0.011 | 0.263±0.019 | 0.204±0.022 | **0.233±0.023** | **0.199±0.009** |
| r=3 | .40 | 0.233±0.010 | 0.150±0.003 | **0.287±0.024** | 0.197±0.014 | 0.297±0.018 | **0.243±0.019** | 0.199±0.007 | 0.152±0.007 | 0.243±0.005 | **0.153±0.005** | 0.254±0.006 | 0.214±0.007 |
| r=4 | .52 | 0.217±0.014 | **0.145±0.003** | 0.294±0.027 | **0.194±0.014** | 0.325±0.013 | 0.288±0.032 | 0.180±0.009 | **0.143±0.006** | **0.236±0.008** | 0.167±0.010 | 0.305±0.010 | 0.307±0.028 |
| r=5 | .62 | 0.226±0.022 | 0.146±0.006 | 0.303±0.012 | 0.200±0.019 | 0.349±0.006 | 0.331±0.019 | **0.165±0.010** | 0.144±0.005 | 0.254±0.015 | 0.175±0.006 | 0.331±0.013 | 0.297±0.023 |
| r=6 | .71 | **0.206±0.005** | 0.169±0.010 | 0.305±0.014 | 0.209±0.016 | 0.373±0.012 | 0.343±0.009 | 0.171±0.007 | 0.152±0.007 | 0.255±0.009 | 0.185±0.009 | 0.352±0.005 | 0.337±0.009 |
| r=7 | .79 | 0.206±0.013 | 0.165±0.008 | 0.318±0.012 | 0.211±0.017 | 0.371±0.017 | 0.336±0.003 | 0.172±0.007 | 0.152±0.004 | 0.259±0.013 | 0.197±0.004 | 0.351±0.005 | 0.327±0.012 |
| r=8 | .85 | 0.212±0.012 | 0.180±0.010 | 0.341±0.035 | 0.235±0.031 | 0.369±0.009 | 0.338±0.009 | 0.192±0.008 | 0.182±0.012 | 0.276±0.019 | 0.210±0.025 | 0.345±0.006 | 0.330±0.006 |
| r=9 | .90 | 0.216±0.007 | 0.203±0.022 | 0.385±0.013 | 0.225±0.009 | 0.396±0.008 | 0.342±0.007 | 0.214±0.012 | 0.257±0.017 | 0.280±0.020 | 0.205±0.011 | 0.363±0.017 | 0.332±0.011 |
| r=10 | .94 | 0.247±0.021 | 0.238±0.014 | 0.366±0.027 | 0.245±0.023 | 0.398±0.009 | 0.350±0.011 | 0.270±0.045 | 0.304±0.032 | 0.275±0.008 | 0.206±0.010 | 0.370±0.013 | 0.336±0.003 |

gcn: GatedGCN, trans: Transformer, adj: adjacency p.e., short: shortest-path p.e., lp: spectral p.e., cls: CLS-node

Table 3: Increasing $r$ on AQSOL 100K parameters.

| type: | density | gcn-adj | gcn-adj-cls | gcn-short | gcn-short-cls | gcn-lp | gcn-lp-cls | mon-adj | mon-adj-cls | mon-short | mon-short-cls | mon-lp | mon-lp-cls |
|---|---|---|---|---|---|---|---|---|---|---|---|---|---|
| r=1 | .17 | 1.277±0.039 | **0.920±0.009** | 1.287±0.017 | **0.927±0.019** | 1.027±0.006 | 0.936±0.004 | 1.391±0.019 | 1.261±0.117 | 1.402±0.013 | 1.216±0.139 | **1.136±0.020** | 1.234±0.028 |
| r=2 | .37 | 1.268±0.011 | 0.956±0.019 | 1.273±0.019 | 0.947±0.016 | 1.049±0.016 | 0.961±0.027 | 1.357±0.020 | 1.205±0.049 | 1.376±0.032 | 1.145±0.055 | 1.193±0.021 | 1.269±0.155 |
| r=3 | .54 | 1.164±0.006 | 0.954±0.013 | 1.200±0.013 | 0.961±0.017 | 1.045±0.010 | 0.953±0.020 | 1.250±0.009 | 1.277±0.088 | 1.269±0.017 | 1.215±0.070 | 1.160±0.023 | **1.183±0.017** |
| r=4 | .67 | 1.118±0.008 | 0.943±0.017 | 1.132±0.012 | 0.951±0.008 | 1.056±0.007 | 0.937±0.019 | 1.240±0.018 | 1.183±0.039 | 1.188±0.027 | 1.158±0.036 | 1.199±0.021 | 1.225±0.056 |
| r=5 | .76 | 1.076±0.015 | 0.970±0.011 | 1.090±0.019 | 0.981±0.012 | 1.046±0.022 | 0.962±0.012 | 1.243±0.040 | **1.166±0.026** | **1.179±0.030** | 1.183±0.071 | 1.211±0.005 | 1.203±0.026 |
| r=6 | .82 | 1.056±0.021 | 0.941±0.020 | 1.064±0.017 | 0.945±0.012 | 1.054±0.018 | 0.933±0.008 | 1.229±0.031 | 1.195±0.041 | 1.206±0.013 | 1.151±0.044 | 1.194±0.014 | 1.217±0.038 |
| r=7 | .87 | 1.064±0.014 | 0.967±0.018 | 1.043±0.012 | 0.949±0.006 | **1.026±0.017** | **0.930±0.004** | 1.235±0.056 | 1.186±0.031 | 1.197±0.024 | **1.141±0.035** | 1.208±0.028 | 1.211±0.022 |
| r=8 | .90 | **1.054±0.014** | 0.956±0.017 | 1.057±0.008 | 0.952±0.009 | 1.035±0.009 | 0.944±0.014 | **1.213±0.025** | 1.179±0.043 | 1.212±0.022 | 1.144±0.016 | 1.193±0.013 | 1.182±0.019 |
| r=9 | .92 | 1.090±0.009 | 0.996±0.009 | 1.042±0.008 | 0.955±0.010 | 1.038±0.011 | 0.952±0.010 | 1.248±0.044 | 1.205±0.055 | 1.205±0.017 | 1.160±0.050 | 1.184±0.014 | 1.208±0.008 |
| r=10 | .93 | 1.092±0.010 | 0.966±0.008 | **1.035±0.009** | 0.953±0.018 | 1.037±0.011 | 0.951±0.010 | 1.228±0.011 | 1.264±0.067 | 1.164±0.037 | 1.161±0.038 | 1.195±0.024 | 1.192±0.015 |

gcn: GatedGCN, mon: MoNet, adj: adjacency p.e, short: shortest-path p.e., lp: spectral p.e., cls: CLS-node

Table 4: Increasing $r$ on CLUSTER 100K parameters.

| type: | density | trans-adj | trans-adj-cls | trans-short | trans-short-cls | trans-lp* | trans-lp-cls* | gcn-adj | gcn-adj-cls | gcn-short | gcn-short-cls | gcn-lp | gcn-lp-cls |
|---|---|---|---|---|---|---|---|---|---|---|---|---|---|
| r=1 | .31 | 73.124±0.264 | 73.972±0.123 | 73.346±0.119 | 74.117±0.363 | **53.858±7.832** | 48.950±6.887 | 72.492±0.460 | 73.459±0.197 | 72.554±0.418 | 73.048±0.220 | 76.453±0.105 | 77.156±0.181 |
| r=2 | >.99 | 76.964±0.059 | 77.193±0.072 | **76.498±0.216** | **76.432±0.115** | 47.140±11.138 | 53.381±4.887 | **76.917±0.059** | **76.874±0.172** | **75.354±0.115** | **75.411±0.063** | 77.445±0.153 | 77.520±0.176 |
| r=3 | 1.0 | **77.095±0.250** | **77.266±0.133** | 76.364±0.085 | 76.636±0.049 | 37.274±14.859 | **54.194±1.746** | 61.028±2.334 | 61.540±2.404 | 75.255±0.199 | 75.392±0.190 | **77.575±0.149** | **77.560±0.195** |

gcn: GatedGCN, trans: Transformer, adj: adjacency p.e., short: shortest-path p.e., lp: spectral p.e., cls: CLS-node, *: Training not converging

Table 5: Increasing $r$ on PATTERN 100K parameters.

| type: | density | gcn-adj | gcn-adj-cls | gcn-lp | gcn-lp-cls | gcn-short | gcn-short-cls |
|---|---|---|---|---|---|---|---|
| r=1 | .43 | 85.715±0.036 | **86.723±0.006** | 86.547±0.026 | 86.713±0.031 | 85.681±0.033 | 86.732±0.020 |
| r=2 | >.99 | **86.698±0.047** | 86.707±0.029 | **86.723±0.031** | **86.747±0.011** | **86.757±0.031** | 86.736±0.014 |
| r=3 | 1.0 | 85.471±0.949 | 84.657±0.977 | 86.718±0.024 | 86.744±0.015 | 86.712±0.031 | **86.739±0.027** |

gcn: GatedGCN, adj: adjacency p.e, short: shortest-path p.e., lp: spectral p.e., cls: CLS-node

Table 6: Increasing $r$ on MNIST 100K parameters.

| type: | density | gcn-adj | gcn-adj-cls | gcn-lp | gcn-lp-cls | gcn-short | gcn-short-cls |
|---|---|---|---|---|---|---|---|
| r=1 | .13 | 98.537±0.089 | 98.522±0.033 | 98.395±0.099 | 98.542±0.079 | 98.373±0.126 | 98.545±0.057 |
| r=2 | .34 | **98.630±0.134** | **98.743±0.062** | **98.720±0.067** | **98.605±0.032** | **98.597±0.070** | **98.552±0.107** |
| r=3 | .58 | 98.035±0.094 | 98.190±0.141 | 98.513±0.145 | 98.570±0.117 | 98.315±0.156 | 98.390±0.104 |

gcn: GatedGCN, adj: adjacency p.e., short: shortest-path p.e., lp: spectral p.e., cls: CLS-node

## 6.1 NeighborsMatch, Over-squashing

Alon and Yahav (2021) introduce a toy problem called *NeighborsMatch* to benchmark the extent of over-squashing in GNNs, while controlling over-squashing by limiting the problem radius $r_p$. The graphs in the dataset are binary trees of depth equal to the problem radius $r_p$. Thus, the graphs are

Table 7: Increasing $r$ on CIFAR10 100K parameters.

| type: | density | gcn-adj | gcn-adj-cls | gcn-lp | gcn-lp-cls | gcn-short | gcn-short-cls |
|---|---|---|---|---|---|---|---|
| r=1 | .08 | **73.415±0.717** | **73.498±0.842** | **72.525±0.471** | **73.808±0.193** | **72.610±0.574** | **72.950±0.520** |
| r=2 | .21 | 72.037±0.400 | 72.480±0.420 | 72.085±0.487 | 71.745±0.325 | 72.127±0.471 | 71.470±0.508 |
| r=3 | .38 | 70.688±0.171 | 69.580±0.488 | 70.380±0.308 | 70.318±0.295 | 71.285±0.722 | 71.188±0.498 |

gcn: GatedGCN, adj: adjacency p.e., short: shortest-path p.e., lp: spectral p.e., cls: CLS-node

structured and sparse, and the number of edges grows linearly with the increased receptive field $r$. See Figure 1, Appendix C, for results with GatedGCN. Increasing the receptive field $r$ with a step of 1 increases the attainable problem radius with a step of 1, while using the CLS-node at $r = 1$ falls in between the performance of $r = 2$ and $r = 3$ but with a much longer tail. Thus, this further showcases the subsumed as well as different effect (complementary and conflicting) the receptive field and the CLS-node have, as also observed on the other benchmarks.

## 6.2 COMPUTATIONAL ANALYSIS

For all positional encodings, the number of edges determines the asymptotic runtime and memory use. The CLS-node only introduces an additive factor. Figures 4 and 5 in Appendix E show that the runtime in practice scales roughly the same as the density, as the receptive field size is increased; though real runtime has a significant constant factor.

## 6.3 SELECTING POSITIONAL ENCODING AND HOPS SIZE

We recommend the adjacency positional encodings together with the CLS-node. In terms of ranked performance across the 6 datasets, adjacency- and spectral positional encodings perform the same, but the spectral encoding performs considerably worse on the ZINC dataset, while the differences are smaller on the other datasets. Additional experiments in Appendix D, Figure 2, assess the discriminative power of the different encodings. However, there is no positional encoding superior in all aspects. Instead, each one has unique benefits as well as drawbacks. This is made apparent by considering $r$ as a parameter and observing the performance differences across values of $r$. Furthermore, the CLS-node is part of the best-performing configuration more often than not. Similarly, no fixed $r$ is optimal for all datasets. Instead, optimal $r$ depends on the dataset and the amount of compute. Appendix F shows that increased $r$ diminishes the reliance on homophily as an inductive bias, and thus low homophily of a dataset could be used as an indicator for selecting an increased $r$. If the density does not change much from a change in $r$ then neither does performance. The use of the spectral positional encodings, the CLS-node, or increased $r$ have subsuming effects for multiple datasets; here the CLS-node or spectral positional encodings may be preferred, computationally cheaper, alternatives to increasing $r$.

From this empirical study, for picking optimal $r$, we recommend computing the densities for increasing $r$ and picking the first one where the average density exceeds 0.5 to reap most of the performance boosts. This seems to maintain a helpful locality bias as well as to significantly reduce the compute compared to complete attention. See Appendix G for further discussion.

## 7 DISCUSSION

Our simple graph rewiring and positional encodings achieve state-of-the-art performance, widening receptive fields while alleviating over-squashing. This is much due to the ability to easily apply our method to models that stem from a large body of work on GNNs, highlighting the benefits of our model-agnostic approach.

The reality is that attention with complete receptive fields is still computationally intractable for most practitioners and researchers. However, here we show that significant performance boosts via attention and increased receptive fields can be obtained by increasing the receptive field only slightly. Thus, opening up recent work to a broader range of practitioners as well as giving more fair conditions for comparing GNNs. In addition, the systematic investigation of increased receptive fields and positional encodings gives further insights into the necessity of homophily for the success of GNNs and highlights other implicit biases in GNN architectures.

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

# A    TRAINING DETAILS

Both code and training follow Dwivedi et al. (2020) closely, and to a lesser extent (Dwivedi and Bresson, 2021), which uses the same code base.

Like (Dwivedi et al., 2020), we use the Adam optimizer (Kingma and Ba, 2015) with the same learning rate decay strategy. The initial learning rate is set to $10^{-3}$ and is reduced by half if the validation loss does not improve after a fixed ("lr_schedule_patience") number of epochs, either 5 or 10. Instead of setting a maximum number of epochs, the training is stopped either when the learning rate has reached $10^{-6}$ or when the computational time reaches 12 hours (6 hours for NeighborsMatch). Experiments are run with 4 different seeds; we report summary statistics from the 4 results.

Below we include training settings for the different datasets.

## A.1    ZINC

```
"model": GatedGCN and Transformer,
"batch_size": 128,
"lr_schedule_patience": 10,
"max_time": 12
```

## A.2    AQSOL

```
"model": GatedGCN and MoNet,
"batch_size": 128,
"lr_schedule_patience": 10,
"max_time": 12
```

## A.3    CLUSTER

```
"model": GatedGCN and Transformer,
"batch_size": 48 (GatedGCN), 32 or 16 (Transformer),
"lr_schedule_patience": 5,
"max_time": 12
```

## A.4    PATTERN

```
"model": GatedGCN,
"batch_size": 48,
"lr_schedule_patience": 5,
"max_time": 12
```

## A.5    MNIST

```
"model": GatedGCN,
"batch_size": 128,
"lr_schedule_patience": 10,
"max_time": 12
```

## A.6    CIFAR10

```
"model": GatedGCN,
"batch_size": 128,
"lr_schedule_patience": 10,
"max_time": 12
```

## A.7    NEIGHBORSMATCH

```
"model": GatedGCN,
```

```
"batch_size": 256,
"lr_schedule_patience": 10,
"max_time": 6
```

## B  TRANSFORMER IMPLEMENTATION

We implemented a simple version of the Transformer adapted to graphs:

$$\hat{h}_i^l = \text{BN}(h_i^{l-1})$$

$$\hat{\hat{h}}_i^l = \big\|_{k=1}^H \Big( \sum_{j \in \mathcal{N}_i \cup \{i\}} a_{i,j}^{l,k} W_k^l \hat{h}_j^{l-1} \Big) + h_i^{l-1}$$

$$h_i^l = \text{FFN}(\text{BN}(\hat{\hat{h}}_i^l)) + \hat{\hat{h}}_i^l$$

with

$$\hat{e}_{i,j}^l = \text{BN}(e_{i,j}^{l-1})$$

$$\hat{a}_{i,j}^{l,k} = ((A_k^l \hat{h}_i^l)^T (B_k^l \hat{h}_j^l) + C_k^l \hat{e}_{i,j}^l)/d$$

$$a_{i,j}^{l,k} = \frac{\exp(\hat{a}_{i,j}^{l,k})}{\sum_{j' \in \mathcal{N}_i \cup \{i\}} \exp(\hat{a}_{i,j'}^{l,k})}$$

$$e_{i,j}^l = \text{FFN}(\hat{e}_{i,j}^l) + e_{i,j}^{l-1}$$

Here, $h$ and $e$ are node and edge features (resp.) from the previous layer. $W_k, A, B \in \mathbb{R}^{d/H \times d}$ and $C \in \mathbb{R}^{1 \times d}$ are learnable weight-matrices, $H$ is the number of attention heads, and BN is short for batch normalization. $\big\|_{k=1}^H$ denotes the concatenation of the attention heads.

### B.1  DESIGN CHOICES AND CHALLENGES

There are many variations on the Transformer model. Following Ying et al. (2021), we put the normalization before the multi-head attention, which caused instability when training on CLUSTER with Laplacian (spectral) positional encodings. This was fixed by putting the normalization after or using layer normalization instead of batch normalization; however, these changes reduced performance on ZINC. While the GatedGCN worked well with identical architecture parameters across datasets, we found that the Transformer needed more variations to stay competitive on MNIST and CIFAR10; in particular, fewer layers and larger hidden dimensions.

Transformers use multi-head attention which puts number-of-heads dimension vectors on each edge—seen as directed. Hence, the memory load becomes $2 \times |E| \times$ num_heads (in our experiments, num_heads = 6), which compared for GatedGCN is only $2 \times |E|$. This causes a memory bottleneck for the Transformer that may force one to use a reduced batch size to avoid memory issues.

### B.2  OTHER VARIANTS

We implemented other variants, including more involved Transformers. As in (Vaswani et al., 2017), we ran the path-integers through sine and cosine functions of different frequencies, and inspired by (Dai et al., 2019; Ke et al., 2020) we implemented a more involved incorporation of relative positions in the multi-head attention (see below); however, we found performance to be comparable.

In natural language processing, the input is a sequence (a line graph) $x = (x_1, \ldots, x_n)$ of text tokens from a vocabulary set $\mathcal{V}$, with each token having a one-hot-encoding $f_{\mathcal{V}} : \mathcal{V} \to [0,1]^{|\mathcal{V}|}$. The word embeddings $E \in \mathbb{R}^{n \times d}$ for $n$ tokens are formed as $E = (W_{embed} f_{\mathcal{V}}(x_i) \mid x_i \in x)$ where $W_{embed} \in \mathbb{R}^{d \times |\mathcal{V}|}$ is a learnable weight matrix.

The original Transformer model used *absolute positional encodings*. This means that we add the positional encoding to the node embedding at the input layer. Consider a positional encoding function $p_e : \mathbb{N}_0 \to \mathbb{R}^d$. Then the first input is

$$h^0 = (W_{embed} f_{\mathcal{V}}(x_i) + p_e(i) \mid i = 1, \ldots, n) = E + U$$

Table 8: Increasing $r$ on ZINC/molecules 100K parameters.

| type: | trans-adj | trans-short-cls |
|---|---|---|
| r=1 | 0.338±0.020 | 0.274±0.021 |
| r=2 | 0.296±0.010 | **0.179±0.011** |
| r=3 | 0.260±0.013 | 0.183±0.018 |
| r=4 | 0.255±0.009 | 0.271±0.036 |
| r=5 | 0.235±0.022 | 0.227±0.026 |
| r=6 | 0.226±0.015 | 0.264±0.042 |
| r=7 | 0.219±0.012 | 0.251±0.039 |
| r=8 | **0.210±0.009** | 0.278±0.026 |
| r=9 | 0.213±0.010 | 0.289±0.042 |
| r=10 | 0.564±0.221 | 0.327±0.023 |

Table 9: Increasing $r$ on CLUSTER 100K parameters.

| type: | trans-adj-cls | trans-adj | trans-short-cls | trans-short |
|---|---|---|---|---|
| r=1 | 74.262±0.188 | 73.445±0.068 | 74.717±0.308 | 72.947±0.123 |
| r=2 | **77.390±0.168** | **77.399±0.200** | **76.771±0.012** | 76.454±0.084 |
| r=3 | 77.216±0.226 | 68.384±7.975 | 76.770±0.094 | **76.521±0.250** |

where $U = (p_e(i) \mid i = 0, \ldots n) \in \mathbb{R}^{n \times d}$. Typically $p_e$ contains sine and cosine functions of different frequencies:

$$p_e(k, 2 \times l) = \sin(k/10000^{(2 \times l)/d})$$

$$p_e(k, 2 \times l + 1) = \cos(k/10000^{(2 \times l + 1)/d})$$

where $k \in \mathbb{N}$ is the position and $l \in \mathbb{N}$ is the dimension. That is, each dimension of the positional encoding corresponds to a sinusoid. The wavelengths form a geometric progression from $2\pi$ to $10000 \times 2\pi$. This function was chosen because it was hypothesized that it would allow the model to easily learn to attend by relative positions, since for any fixed offset $m$, $p_e(k + m)$ is a linear function of $p_e(k)$. It was also hypothesized that it may allow the model to extrapolate to sequence lengths longer than the ones encountered during training.

In many cases, absolute positional encodings have been replaced with *relative fully learnable positional encodings* and *relative partially learnable positional encodings* (Dai et al., 2019). To justify these, consider the first attention layer with absolute positional encodings:

$$A_{i,j}^{abs} = E_{x_i} W_q W_k^T E_{x_j}^T + E_{x_i} W_q W_k^T U_j^T + U_i W_q W_k^T E_{x_j}^T + U_i W_q W_k^T U_j^T$$

For relative (fully and partially) learnable positional encodings we have instead:

$$A_{i,j}^{rel} = E_{x_i} W_q W_{k,E}^T E_{x_j}^T + E_{x_i} W_q W_{k,R}^T R_{i-j}^T + u W_{k,E}^T E_{x_j}^T + v W_{k,R}^T R_{i-j}^T$$

where $u, v \in \mathbb{R}^{1 \times d}$ are learnable weights and $R_{i-j} \in \mathbb{R}^{1 \times d}$ is a relative positional encoding between $i$ and $j$. Each term has the following intuitive meaning: term (1) represents content-based addressing, term (2) captures a content-dependent positional bias, term (3) governs a global content bias, and (4) encodes a global positional bias.

For relative fully learnable positional encodings, $W_{k,R}^T R_{i-j}^T$ is a learnable weight in $\mathbb{R}^{d \times 1}$ for each $i - j \in \mathbb{N}$, while for relative partially learnable positional encodings $R_{i,j} = p_e(|i - j|)$ where $p_e$ is the sinusoidal function from before.

We implemented both fully and partially learnable positional encodings for the shortest-path positional encodings (integer-valued) and related versions for the other positional encodings (in $\mathbb{R}^d$). We include results in Tables 8 and 9.

## C    OVER-SQUASHING

Results for over-squashing experiment can be found in Figure 1.

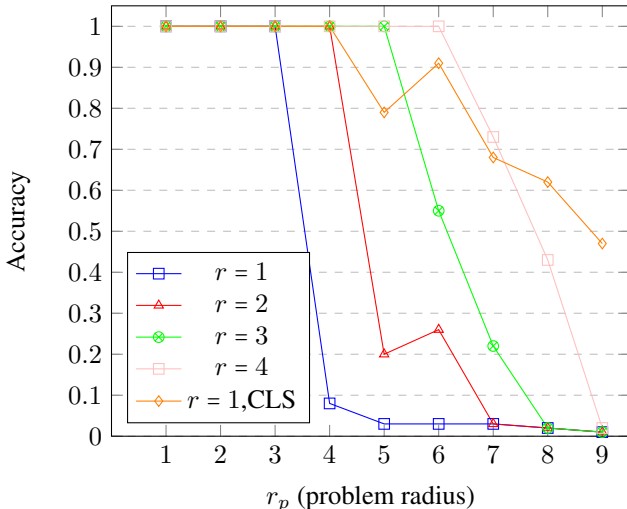

Figure 1: NeighborsMatch (Alon and Yahav, 2021). Benchmarking the extent of over-squashing via the problem radius $r_p$.

## D ADDITIONAL EVALUATION OF POSITIONAL ENCODINGS

Here we provide a start to toy data and a task for comparing positional encodings. In this task we wish to assess how powerful the positional encodings are in practice, i.e. how well they discriminate between different graph isomorphism classes. Specifically, we generate 100 random Erdos graphs and then expand the receptive field so that the graph is fully connected. Thus, the positional encodings become the mere instrument for communicating the connectivity/topology of the original graph. The task is to retrieve a specific graph among all 100 graphs, i.e. the task is graph classification and there is a 100 classes. Hence, achieving 100% accuracy means that the GNN, based on the positional encodings, has been able to discriminate between all graphs. We only look at train accuracy here, since we're interested in the power to overfit, not to generalize. Results can be found in Figure 2.

All positional encodings are able to solve that task after a sufficient amount of training, besides Adj-10. Adj-5 and Adj-10 encode the adjacency matrix to the power of 5 and 10 respectively (at both points all graphs are fully connected). Adj-10 encodes between any two nodes the number of paths of length 10, number of path of length 9, and so on. The experiments indicate that too much such information confuses the GNN and makes it harder to discriminate between graphs. The shortest and Adj-5 positional encodings are the fastest at solving the task. This can be due to the fact that the Laplacian positional encoding is only unique up to a sign and that we randomly switch the sign during training.

## E COMPUTATIONAL RUNTIME AND MEMORY USE

In our implementation, the step of computing the positional encodings as well as expanding the r-hops of the graph is done in the same process for shortest-path and adjacency positional encodings; thus this step always occur and we found that implementing it via iterative matrix multiplications of the adjacency matrix gave the fastest results. How this scales with the $r$-size can be found in Figure 3. Since each increment of the $r$-size results in an additional matrix multiplication, the linear increase is expected. The spectral positional encoding has the same additive runtime per graph across $r$-sizes of $1.3 \times 10^{-3}$ seconds. These matrix multiplications are done on CPU rather than GPUs, but running them on GPUs could results in speed-ups. However, the runtime for computing these positional encodings is at least an order of magnitude smaller (per graph) than the runtime for running the subsequent GNN on a GPU, so there was no need to optimize this runtime further.

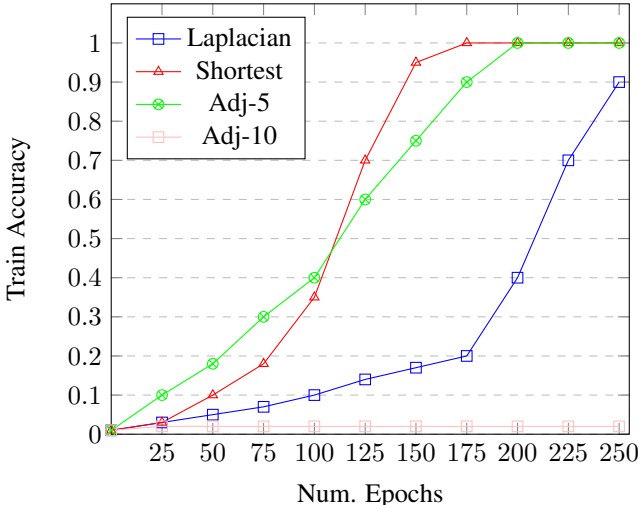

Figure 2: Learning to retrieve random Erdos graphs.

In Figures 4 and 5 we include actual runtime of the GNN (on GPU) of different positional encodings and hops sizes, juxtaposed with the density of the modified graphs, for the ZINC and CIFAR10 datasets. Note, we are here excluding the computation of the positional encoding on the input graph, which can be found in Figure 3.

Most graphs to which GNNs are applied to are connected and typically the number of edges are greater than the number of nodes, i.e. $|E| \geq |V|$. Since all established GNNs make use of the edges in one way or another, the number of edges usually determines the asymptotic behavior of the runtime and memory use, i.e. they are in $O(|E|)$. With modern deep learning and specialized graph learning framework, GPU-parallelization and other more technical aspect affect memory and runtime. Thus, Figures 4 and 5 compare theoretical runtime (dominated by the density) with actual runtime of code run on GPUs. We find that density and actual runtime is strongly correlated. In Figure 6 we include the memory use for increasing radius on ZINC dataset, and find its roughly linear with the density as well.

## F   HOMOPHILY SCORE AND PERFORMANCE

We include experiments to investigate the correlation between homophily score (Ma et al., 2021) and performance when increasing hops size. This applies to the node classification datasets, CLUSTER and PATTERN, that we used. We split the test set into three buckets, which is just a sorted segmentation of the graphs with increasing homology scores. We evaluate trained Gated-GCN models with adjacency positional encodings for $r$-values 1 and 2 (at $r = 2$ almost all graphs are fully connected). See Tables 10 and 11 for results.

We find that high homophily score correlates much stronger with performance when $r = 1$ than it does at $r = 2$. This indicates that increased r-size diminishes the reliance on homophily as an inductive bias.

Table 10: Increasing $r$ on CLUSTER homophily buckets.

| density | homophily score: | 0.315±0.008 | 0.336±0.006 | 0.366±0.0160 |
|---------|------------------|-------------|-------------|--------------|
| .31     | r=1              | 71.494±0.619 | 72.361±0.462 | 73.812±0.265 |
| >.99    | r=2              | 77.010±0.234 | 76.660± 0.143 | 76.965±0.242 |

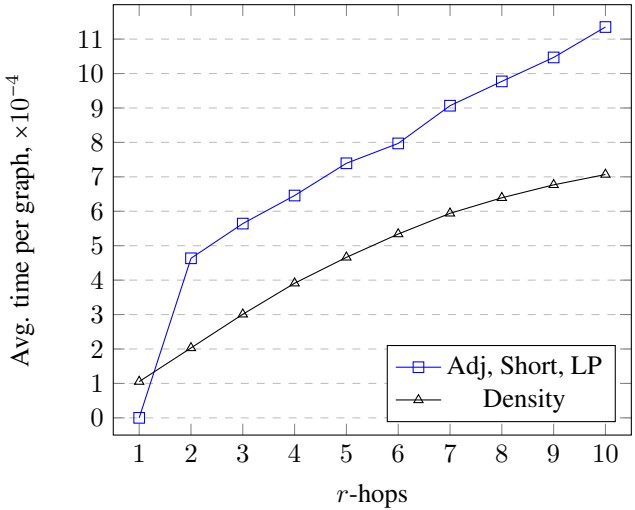

Figure 3: P.E. Computational Time on ZINC/molecules

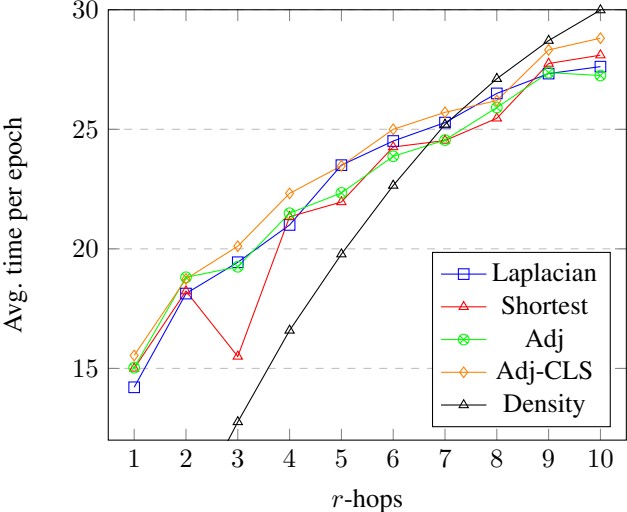

Figure 4: GNN Computational Time on ZINC/molecules

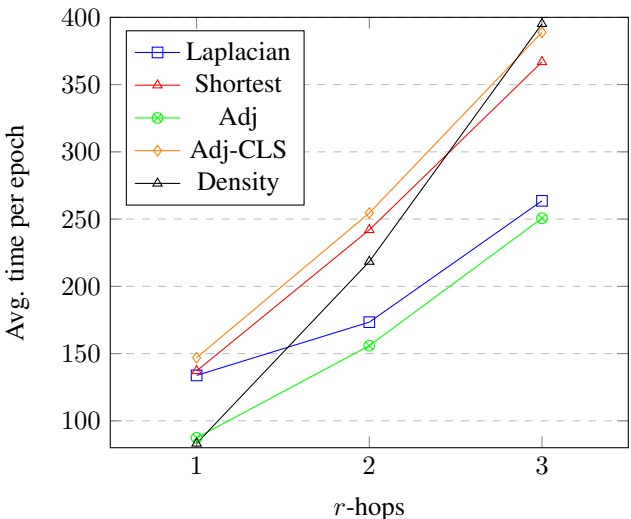

Figure 5: GNN Computational Time on CIFAR10

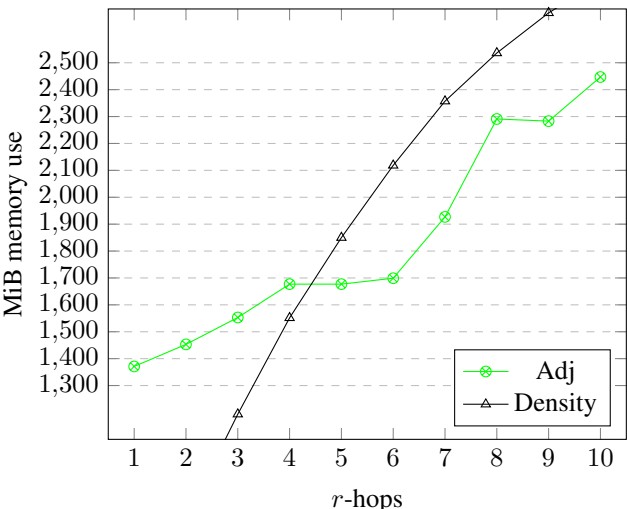

Figure 6: GNN Memory use on ZINC/molecules

Table 11: Increasing $r$ on PATTERN homophily buckets.

| density | homophily score: | 0.567±0.031 | 0.652±0.026 | 0.774±0.051 |
|---------|------------------|-------------|-------------|-------------|
| .43 | r=1 | 83.958±0.031 | 86.862±0.021 | 92.154±0.492 |
| >.99 | r=2 | 84.197±0.048 | 87.214± 0.062 | 87.085±0.082 |

## G  OPTIMAL POSITIONAL ENCODING AND HOPS SIZE

Again, we recommend the adjacency positional encodings together with the CLS-node. We find that in terms of ranked performance on the 6 datasets, adjacency- and spectral positional encodings perform at the same level, but the spectral encoding perform considerably worse on the ZINC datasets, while the differences are smaller on the other datasets. The spectral encoding hardcode global-to-local information on the nodes and the size of the encoding-vector is a hyper parameter; we found performance not to be too sensitive to this hyper-parameter but future work could further investigate this. Spectral embeddings also use less memory as it does not encode its embeddings as edge-features; however, since information still is propagated along edges we find this memory saving to be significant but not asymptotically different. Adjacency encoding breaks down faster as the $r$-size is increased compare to the other positional encodings, we believe this to be due to the corresponding increase in size of the embedding-vectors and its introducing low-signal information that is also easy to overfit to, e.g. the number of paths of length 10 between two nodes (where any edge can be used multiple times). The Erdos experiments in Appendix D support this observation. However, all in all, the adjacency encoding stands out slightly considering the performance, runtime, memory use, and toy experiments. Furthermore, the CLS-node is part of the best performing configuration more times than it is not, and it has the additional advantage of leading to peak performance at lower $r$-sizes where in some cases it also has reduced runtime and memory use compared instead to increasing the $r$-size.

In this work we do not find a fixed $r$-size that is optimal for all datasets. The optimal $r$ depends on the dataset and the amount of compute available. Given the fixed amount of compute used in our experiments, we found that all the best performance was found at $r$-size four or smaller. We provide heuristic for selecting a good $r$-size but ultimately it depends on the amount of compute and memory available.

