# OpenReview forum: "Rewiring with Positional Encodings for GNNs"
_ICLR.cc/2023/Conference — Submitted to ICLR 2023_

### Official Review · Reviewer_37EK · 2022-10-19

**Confidence:** 4
**Correctness:** 2
**Technical Novelty And Significance:** 2
**Empirical Novelty And Significance:** 2
**Recommendation:** 5

**Clarity, Quality, Novelty And Reproducibility:**

\
**Clarity**

The paper is generally well-organised and well-written with a few caveats.

In the introductory part of Section 4 (titled Approach), the authors abuse the notation $\mathcal{G}$ slightly and then use $\mathcal{G}$ to denote only the subset of graphs relevant to a given machine learning (ML) problem (e.g., molecular graphs).

It is unclear how relevant the rewired graph given by $g:\mathcal{G}\rightarrow\mathcal{G}$ would be for the ML problem (i.e, it is unclear if the rewired graph would still be a molecule and if so how relevant would it be).

\
**Quality**

The quality of the paper can be strengthened regarding the arguments around the choice of $r$ (i.e., the hop size) and the homophily ratio.

The authors investigate correlation between homophily score and performance for increasing $r$ on SBM datasets (CLUSTER and PATTERN).

It would be much more compelling to investigate the correlation on real-world node classification datasets with low homophily [1] and high homophily [2].

1. Large Scale Learning on Non-Homophilous Graphs: New Benchmarks and Strong Simple Methods, In NeurIPS'21,
2. Open Graph Benchmark: Datasets for Machine Learning on Graphs, In NeurIPS'20.



\
**Novelty**

The novelty of the work can be strengthened by discussing existing work on using virtual/dummy nodes and positional encodings to boost GNNs.
* Boosting Graph Structure Learning with Dummy Nodes, In ICML'22,
* Graph Neural Networks with Learnable Structural and Positional Representations, In ICLR'22,
* Equivariant and Stable Positional Encoding for More Powerful Graph Neural Networks, In ICLR'22.


\
**Reproducibility**

The code is not provided although the main part and the appendix include enough material, e.g., dataset details, baselines with references, hyperparameters, for an expert to replicate the results of the paper.

___

**Strength And Weaknesses:**

\
**Strengths**

\+ The proposed methods are evaluated on node classification, graph classification, regression tasks (two datasets each) and the NeighboursMatch problem (for oversquashing) and compared with seven baseline models.

\+ Based on the empirical evaluations of different $r$ values with and without virtual nodes and three different positional encodings, the authors conclude that the adjacency positional encoding with virtual nodes works best and an optimal $r$ (i.e., hop size) generally depends on the homophily of the dataset.


\
**Weaknesses**

\- The node classification datasets (PATTERN, CLUSTER) are generated using the stochastic block model (i.e., synthetic in nature).

\- The ideas of using positional encodings (PEs) for GNNs on molecular graph regression is not new, see for instance a prior work [Graph Neural Networks with Learnable Structural and Positional Representations, In ICLR'22].

\- The idea of using a virtual node connecting all existing graph nodes without affecting the graph topology (i.e., ensuring that there is an inverse map back to the original graph) is also not new, see for instance a relevant prior work [Boosting Graph Structure Learning with Dummy Nodes, In ICML'22].

___

**Summary Of The Paper:**

Recent efforts towards addressing under-reaching and over-squashing in graph neural networks (GNNs) include graph transformers (GTs) with topology-based positional encodings; however, the computational cost is quadratic in nature (since vanilla GTs act on fully connected graphs).

This paper proposes and studies a model-agnostic graph rewiring approach with
* edges added between nodes and their r-hop neighbours (the paper studies different values of $r$ from 1 to the graph diameter) and
* a virtual node connecting all the other nodes of the original graph (to encode global graph information).

To retain the graph toplogical information, the paper also empirically studies three types of positional encodings, viz., (i) shortest path, (ii) Laplacian eigenvectors, (iii) adjacency powers, in the form of node/edge features and shows that GNNs acting on the rewired graph + positional encodings are more effective than traditional baselines (even for small values of $r$) on several datasets.

___

**Summary Of The Review:**

While the proposed methods are evaluated on several datasets with several baselines, the paper can be strenghtened by positioning with relevant existing prior work and more empirical evaluation on real-world data.

___

---

### Official Review · Reviewer_CvhJ · 2022-10-24

**Confidence:** 4
**Correctness:** 3
**Technical Novelty And Significance:** 2
**Empirical Novelty And Significance:** 2
**Recommendation:** 3

**Clarity, Quality, Novelty And Reproducibility:**

This framework is of average quality and kind of lacking innovation. The description of this framework is not very clear and the theoretical analysis is sufficient. In the experimental part, the validity analysis of the results is relatively redundant. The originality of the work is marginally below the average level.

**Strength And Weaknesses:**

Strength
1. The intrinsic design idea and logic of this framework are straightforward and clear, and with sufficient analysis and summary of related works, derivation, and proof, the rationality and validity of the framework are verified theoretically.
2. The experimental settings are very detailed, and the experimental results can demonstrate the effectiveness and superiority of this method.

Weaknesses
1. In my opinion, the biggest problem in your framework is the lack of innovation. The positional coding method and virtual node strategy used in your method are existing methods in the existing work. It can be concluded that your work is to expand the perception threshold from 1-hop neighbor to r-hop neighbor and analyze its effect, which lacks innovation and originality.
2. Though your experiment setup is very complete and the experiment data is very informative, your experiment dataset is kind of small compared to the real-world ones. Try to evaluate it on a larger dataset. Also, why not have some well-known position-encoding based GNNs as your baseline? Such as P-GNN (Position-aware graph neural networks, ICML 2019.) and Graphormer (Do transformers really perform badly for graph representation, NIPS 2021.) There is also a recent work that incorporates position encodings for graph rewiring (Position-aware Structure Learning for Graph Topology-imbalance by Relieving Under-reaching and Over-squashing, CIKM 2022), which also should be included in your baselines.
3. In the main text part, it is best to use a diagram to explain your method design. In addition, in the experimental section of the main text, more interesting experimental results should be emphasized and further analysis should be given. Moreover, a large part of the main text introduces or summarizes other works, and the original work is relatively few.


**Summary Of The Paper:**

This paper proposes a method to augment the input graph with additional nodes/edges and use positional encodings as the node and/or edge features, expanding receptive fields from 1-ring neighborhoods to r-ring neighborhoods. Empirical experiments show that relatively small r-hop neighborhoods sufficiently increase performance across models and that performance degrades in the fully connected setting.

**Summary Of The Review:**

The authors propose a method to augment the input graph with additional nodes/edges and use positional encodings as the node and/or edge features, expanding receptive fields from 1-ring neighborhoods to r-ring neighborhoods. However, this work lacks some innovation and has obvious limitations and deficiencies in the experimental datasets and baselines.

---

### Official Review · Reviewer_qbsV · 2022-10-25

**Confidence:** 5
**Correctness:** 4
**Technical Novelty And Significance:** 3
**Empirical Novelty And Significance:** 3
**Recommendation:** 3

**Clarity, Quality, Novelty And Reproducibility:**

* Clarity. The paper well-written and easy to follow.
* Quality. Can be considered as an "empirical" paper whose objective is to elucidate the role of transitivity in certain types of graphs.
* Novelty. Moderate (incremental wrt positional encodings in Transformers).
* Reproducibility. Code follows the Transformer Implementation. No code was released.

**Details Of Ethics Concerns:**

Ok

**Strength And Weaknesses:**

* Strength: Model agnosticism is interesting,
* Weaknesses: As stated above, the approach is neither inductive (positional encodings must be computed for every input graph in graph classification) nor scalable (e.g. the computation of transitive information even when r=1,2 may be prohibitive). The number of experimental baselines is very limited.

**Summary Of The Paper:**

Main concern. A catalog of experiments to find the best combination of positional encoders for any explored type of graphs. The most successful encoding (powers of adjacency matrices) is not scalable unless large graphs are previously sampled. The argument that the powers of adjacencies for small r=1,2 beats full attention is true, but this is not enough to validate the approach.

I agree with the model agnosticism of the proposal but it is not inductive at all (positional encodings must be computed for any input graph in the case of graph classification).

POSITIONAL ENCODINGS: ONLY TESTED WITH SBMs and KNN-GRID graphs. The conclusions obtained for spectral positional encoding match the intuition and can be expected by the analysis of the type of graph. In any case, incorporating the above analysis makes the paper less “empirical” and more “principled”.

a) In SBM; graphs is logical that spectral positional encoding do not work well for r>1 because the structure of the SBM is encoded by the first  K non-trivial Laplacian eigenvalues if there are K communities (depending on the inter-class structural noise). For instance, a nice experiment should be to track the performance of the positional encoders as the inter-class structural noise increases. In the experiments in the paper one can infer that if the spectral info is not useless for r<3 in CLUSTER/PATTERN is because the interclass structural noise is large

b) On the other hand, KNN graphs (e.g. CIFAR) or GRID graphs (MNIST) are typically broken in two communities just using the first non-trivial eigenvector (Fiedler vector). For this analysis see “SPECTRE: Spectral Conditioning Helps to Overcome the Expressivity Limits of One-shot Graph Generators”.

Regarding the success of adjacency powers (which generalize shortest paths), it is interesting to confirm how not too many hops are needed in general. I suggest addressing these powers in terms of how powerful are to encode different orders of transitive information. However, the main drawback is that this method does not scale well in real-life graphs (e.g. reported out-of-memory in MNIST).

There are no experiments on SOCIAL NETWORKS (e.g. power-law) where hubs do exist and incorporate naturally the CLS-node concept. They also make shortest paths almost uniform (unit length), E.g. in the case of Facebook (“friends circles”) with two-step (r=2) separation, the proposed positional encoders become more ambiguous.


**Summary Of The Review:**

A quite empirical paper that selects the best way of  "static" (not inductive) "topological rewiring" depending on the type of graphs. No theoretical insights are given beside a shallow spectral analysis. An important experimental limitation is that no social network is analyzed. The most challenging rewiring strategy (adjacency powers) is not scalable (see also the plots in the appendix).

---

### Official Review · Reviewer_apU4 · 2022-10-25

**Confidence:** 4
**Correctness:** 3
**Technical Novelty And Significance:** 2
**Empirical Novelty And Significance:** 2
**Recommendation:** 5

**Clarity, Quality, Novelty And Reproducibility:**

The work is presented clearly and is easy to follow. The experimental results are thorough and should not be difficult to reproduce. However, The technical novelty of the paper is limited compared to previous works in multi-hop GNNs and positioning encoding of GNNs.

**Strength And Weaknesses:**

Pros:

The detailed experimental results in Section 6 are appreciated.

A thorough analysis of runtime and memory consumption is presented in the appendix which is helpful.


Cons:

The technical novelty of the paper is limited compared to previous works in multi-hop GNNs and positioning encoding of GNNs. Two pages are used to discuss position encodings in Section 4.2. However, it is not clear what is the novel contribution of this work on position encodings. Since this work does not propose a new position encoding, I recommend the authors compress Section 4.2 and move some content to related work.

It is not clear how the results of Table 1 are obtained in section 6. What value of the hyperparameter $r$ and what GNN models are used? Is $r$ the same across datasets or is it tuned for each dataset? The results seem to be aggregated from the best models in Table 2-6. If this is the case, the hyperparameter $r$ are GNN layers are chosen on the test set results with heavy hyperparameter tuning which is not encouraged. Different $r$ and with/without CLS nodes are used in different datasets which also makes in results inconclusive.

This work is a combination of multi-hop GNN methods and positioning encoding methods on graphs. I believe a comparison with multi-hop GNNs is necessary such as MixHop.

Minor:

"Section 4: we also add a fully-connected CLS node." "CLS" node should be clearly defined here.

[1] Abu-El-Haija, S., Perozzi, B., Kapoor, A., Alipourfard, N., Lerman, K., Harutyunyan, H., Ver Steeg, G. and Galstyan, A., 2019, May. Mixhop: Higher-order graph convolutional architectures via sparsified neighborhood mixing. In international conference on machine learning (pp. 21-29). PMLR.

**Summary Of The Paper:**

This paper proposes a method to enlarge the receptive field of GNNs by augmenting graphs with r-hop neighborhoods, positional encodings and classification node. Extensive experiments are conducted on six benchmark graph datasets including ZINC, AQSOL, PATTERN, CLUSTER, MNIST, and CIFAR10 to demonstrate the effectiveness of the proposed method.

**Summary Of The Review:**

My main concerns are the limited novelty of this work compared to previous works in multi-hop GNNs and the positioning encoding of GNNs and the inconclusive evaluations on the graph benchmark with heavy hyperparameters and GNN design choices tunings.

---

### Decision · Program_Chairs · 2023-01-20

**Decision:**

Reject

**Justification For Why Not Higher Score:**

There is a general consensus between all reviewers that major work is needed before this work would pass the bar for ICLR. The authors do not appear to contest this decision.

**Justification For Why Not Lower Score:**

N/A

**Metareview: Summary, Strengths And Weaknesses:**

This paper proposes a method for graph rewiring using positional embeddings. The authors demonstrate some useful outcomes w.r.t. alleviating the oversquashing problem, and the experimental section has generally been seen as thoroughly written up. However, all reviewers pointed out significant limitations in the claims of the work's novelty, the relevance of its experimental setup, and the overall scalability of the method. Since the authors did not provide a rebuttal, I am assuming they also agree with this assessment. I can only recommend rejection in the current form, but I highly recommend the authors to continue improving their work going forward, it has the potential to be valuable. Additionally, I would _strongly_ recommend to the authors to add a figure describing their method in the core body of the paper. Currently the work has no figures in the main content, significantly limiting its readability.